# Electronic Structure and Lithium Diffusion in LiAl_2_(OH)_6_Cl Studied by First Principle Calculations

**DOI:** 10.3390/molecules24142667

**Published:** 2019-07-23

**Authors:** Yueping Zhang, Xiyue Cheng, Chen Wu, Jürgen Köhler, Shuiquan Deng

**Affiliations:** 1State Key Laboratory of Structural Chemistry, Fujian Institute of Research on the Structure of Matter (FJIRSM) Chinese Academy of Sciences (CAS), Fuzhou 350002, China; 2Max-Planck-Institute for Solid State Research, Heisenbergstr. 1, D-70569 Stuttgart, Germany

**Keywords:** layered double hydroxides, LiAl_2_(OH)_6_Cl, ab initio molecular dynamics (AIMD) simulations, superionic conductor

## Abstract

First-principles calculations based on the density functional theory (DFT) were carried out to study the atomic structure and electronic structure of LiAl_2_(OH)_6_Cl, the only material in the layered double hydroxide family in which delithiation was found to occur. Ab initio molecular dynamics (AIMD) simulations were used to explore the evolution of the structure of LiAl_2_(OH)_6_Cl during a thermally induced delithiation process. The simulations show that this process occurs due to the drastic dynamics of Li^+^ at temperatures higher than ~450 K, in which the [Al_2_(OH)_6_] host layers remain stable up to 1100 K. The calculated large value of the Li^+^ diffusion coefficient D, $\sim 3.13\times 10^{-5}cm^{2}/s$, at 500 K and the high stability of the [Al_2_(OH)_6_] framework suggest a potential technical application of the partially-delithiated Li_1-x_Al_2_(OH)_6_Cl_1-x_ (0 < x < 1) as a superionic conductor at high temperatures.

## 1. Introduction

Layered double hydroxides (LDHs) are found among clay minerals or as synthesized materials, which are composed of positively charged brucite-like layers and anions between these layers. Extensive studies show that LDHs have important applications in catalysis [1,2,3,4], biomedical science [5,6], energy science [7,8], etc. Generally, the chemical formula of LDHs can be written as [M^2+^_1−*x*_M^3+^*_x_*(OH)_2_]*^x^*^+^(X*^n^*^−^)*_x_*_/*n*_·*m*H_2_O (X*^n^*^−^ = Cl^−^, NO_3_^−^, Br^−^…) (short form M^2+^M^3+^-X-*m*H_2_O). The water molecules and the anions (X*^n^*^−^) were intercalated between the [M^2+^_1−*x*_M^3+^*_x_*(OH)_2_]*^x^*^+^ layers [9,10,11]. Apart from the common cation combinations of the M^2+^-M^3+^ LDHs, the M^2+^-M^4+^ and the M^+^-M^3+^ LDHs were also reported [12,13,14]. The Li-Al LDHs (LiAl-X-*m*H_2_O), in which the octahedral vacancies of the Al(OH)_3_ layers are completely occupied by monovalent Li^+^ cations [12,13,14,15], are the only known cases of M^+^-M^3+^ LDHs to date.

In the past years, the anion exchange of Li-Al LDHs was extensively studied with respect to their properties and chemical stability. Dutta et al. [12] found that the anion exchange selectivity of the dehydrate LiAl-X (X = PO_4_^3−^, HPO_4_^2−^, Cl^−^, H_2_PO_4_^−^) should follow the order PO_4_^3−^ > HPO_4_^2−^ > Cl^−^ > H_2_PO_4_^−^ by spectroscopic studies and free energy calculations. Fogg et al. [16] applied time-resolved in situ X-ray diffraction (XRD) measurements and obtained that the reaction rate constant into gibbsite Al(OH)_3_ decreases in the series OH^−^ > SO_4_^2−^ > Cl^−^ > Br^−^ > NO_3_^−^. Lei et al. [17,18] studied the preferential anion intercalation of pyridine carboxylate and toluate isomers in LiAl-Cl-*m*H_2_O at different temperatures by nuclear magnetic resonance (NMR) spectra. Hou et al. [19] carried out XRD and NMR to study the hydration state and the factors controlling the expansion behaviors of Li-Al LDHs and Mg-Al LDHs.

In a theoretical approach, Fogg et al. [20] used molecular dynamics to rationalize the structures of LiAl-X (Br^−^, Cl^−^, NO_3_^−^, CO_3_^2−^, SO_4_^2−^, C_2_O_4_^2−^) and showed that this approach is a useful predictive tool for simulating interlayer spacings and the orientations of the guest ions in the related intercalates. The density functional theory (DFT) calculations were performed to study the structural stability of LiAl-X (X = F^−^, Cl^−^, Br^−^, OH^−^, NO_3_^−^, CO_3_^2−^, SO_4_^2−^) and also the CO_2_ capture capacity of LiAl-X (X = Cl^−^, NO_3_^−^, CO_3_^2−^) [21]. In a later study, Cl^−^ position changes were observed in LiAl-Cl by molecular dynamics simulation between −90 and 90 °C [22].

Among the Li-Al LDHs, LiAl-Cl-*m*H_2_O is the only compound in which a delithiation occurs [23]. However, the mechanism of this delithiation (~570 K), after the dehydration (~400 K), and the structural evolution of LiAl-Cl at higher temperatures has not been determined. In particular, the consequence of the delithiation has not been studied, which, however, is rather interesting since the rise of the diffusion rate of the Li^+^ may be due to the increase of the number of vacant Li sites in the structure. A high migrant rate of Li^+^ and a relatively stable structure is the prerequisite condition of an Li-ion battery material. In this paper, we applied the first principle calculations based on DFT to study the electronic structure and bonding of LiAl_2_(OH)_6_Cl. Besides, ab initio molecular dynamics (AIMD) simulations were performed to investigate the structural evolution at higher temperatures. The diffusion of the Li from the cation layer into the inter-layer space and the resulting high Li mobility followed by a further delithiation from the system were further investigated.

## 2. Results and Discussion

### 2.1. Atomic and Electronic Structure of LiAl_2_(OH)_6_Cl

The crystal structures of LiAl-X-*m*H_2_O (X = Cl^−^, NO_3_^−^, Br^−^) and of their dehydrates, LiAl-X, were determined by Besserguenev et al. [15]. In the dehydrates LiAl-X, Li and X form Li⋯X⋯Li⋯X⋯ chains along the c-axis, while the inter chain X⋯X distance is increased in LiAl-X-*m*H_2_O due to the presence of interlayer water molecules [20,21]. Fogg et al. [23,24] found that, in contrast to the rhombohedral modification, the hexagonal LiAl-Cl-*m*H_2_O can form a second-stage intermediate phase and can be isolated as pure crystalline phases during the intercalation reactions with some dicarboxylate anions.

LiAl_2_(OH)_6_Cl crystallizes in the hexagonal space group *P*6_3_/*mcm* (Figure 1a). The optimized lattice parameters are a = 5.158 Å, c = 14.259 Å, which agree quite well with the experimental ones (a = 5.10 Å, c = 14.2994 Å) [15]. The cell parameters and atomic positions obtained from our calculations are all given in Appendix A in the supporting information. Within each unit cell the Li, Al, Cl, O, and H atoms occupy five inequivalent atomic sites, respectively. The Cl atoms are located on a mirror plane between two neighboring [LiAl_2_(OH)_6_]^+^ layers forming Cl-centered trigonal prisms with the six nearest H atoms with Cl–H distances of 2.30 Å (Figure 1a). The Cl–Li distances are much longer (3.56 Å), indicating that the Cl atoms are mainly stabilized by bonding to the H atoms of the hydroxyl groups. As a result, the H atoms of the six hydroxyl groups point towards the Cl atoms in contrast to the situation found in Al(OH)_3_ (Appendix A). As shown in Figure 1b, both the Al and Li atoms are coordinated with six O atoms forming distorted AlO_6_ and LiO_6_ octahedra, respectively. The H atoms are bonded to the O atoms forming hydroxyl groups, with distances of 0.979 Å. Note that the bond lengths of Al–O (1.907 Å) and Li–O (2.132 Å) within each octahedra are equal, while the bond angles of ∠O–Al–O and ∠O–Li–O range from 77.3° to 96.1° and 79.3° and 100.6°, respectively. Therefore, the LiO_6_ octahedra are slightly more distorted than the AlO_6_ octahedra. Besides, each LiO_6_ octahedra shares edges with six AlO_6_ octahedra, which together with the H atoms form positively charged [LiAl_2_(OH)_6_]^+^ layers, see Figure 1b.

The calculated band structures, density of states (DOSs) and crystal orbital Hamilton population (COHP) of LiAl_2_(OH)_6_Cl are shown in Figure 2. The valence band maximum (VBM) is at the K point, while the conduction band minimum (CBM) is located at the Γ point. According to this calculation, LiAl_2_(OH)_6_Cl is a wide-gap insulator with an indirect band gap. The top valence bands are dominated by Cl 3p and O 2p orbitals (~98%) and exhibit small dispersion, as shown in Appendix A. Besides, the composition of the lowest unoccupied bands at the Γ point are made up of Cl 3s and O 2s orbitals (~59.2%) of an antibonding nature. The contribution of the other orbitals to the lowest unoccupied bands at the Γ point are relatively low, with Li 2s (~3.9%), Al 3s (~13.9%), O 2p (~9.2%), H 1s (~4.6%), and H 2p (~9.2%) orbitals.

The bonding properties were analyzed using COHP curves as shown in Figure 2c. Bonding states were mainly observed in the valence band region from –8 to 0 eV for five different kinds of interactions (Li–O, Al–O, H–O, Cl–O, Cl–H). Near E_F_, the H–O interactions show weak antibonding character, while for the Cl–O and Cl–H cases, small bonding states are found. This fact evidences that the interlayer interactions in [LiAl_2_(OH)_6_] via the Cl atoms have mainly an H∙∙∙Cl∙∙∙H bonding character and as a consequence of this weak bonding via H atoms, a mobility of the Cl atoms can be expected.

### 2.2. Lithium Diffusion in LiAl_2_(OH)_6_Cl during Heating

The structural evolution of LiAl_2_(OH)_6_Cl was studied by AIMD simulations at simulated temperatures ranging from 300 K to 1400 K with an increment of 50 K. To monitor the changes of the structure with increasing temperature, we calculated the pair distribution function (PDF), $g_{ij}\left(r\right)$, which gives the probability of finding a *j*-type atom at a distance, *r*, from an *i*-type atom [24,25]. The $g_{ij}\left(r\right)$ is defined as:
(1)$$g_{ij}\left(r\right)=\frac{V}{4\pi r^{2}N_{i}N_{j}}\sum_{i=1}^{N_{i}}\sum_{j=1}^{N_{j}}\delta \left(r_{ij}-r\right),$$
where *V* is the volume of the system, and *N*_i_ and *N*_j_ are the numbers of the *i*-type and *j*-type atoms, respectively. $r_{ij}$ is the length of a vector connecting atom *i* and atom *j*. Figure 3 shows the PDF of Li–O and Al–O at various temperatures. The three relatively sharp peaks for both $g_{LiO}\left(r\right)$ and $g_{AlO}\left(r\right)$ at 400 K correspond to the atomic distances between the nearest, second, and third nearest neighbors of O atoms around Li and Al, respectively, in LiAl_2_(OH)_6_Cl, as expected. All three peaks of both $g_{LiO}\left(r\right)$ and $g_{AlO}\left(r\right)$ get broader in peak width and lower in intensity with increasing temperature, indicating that the Li–O and Al–O interactions become weaker at high temperatures. Due to the lower charge of Li compared to Al, the changes for $g_{LiO}\left(r\right)$ are much larger than for $g_{AlO}\left(r\right)$. For $g_{AlO}\left(r\right)$, the peak positions remain nearly unchanged in our simulations. In contrast, the situation is different for $g_{LiO}\left(r\right)$, where the Li–O distance of around ~2.0 Å, indicated by the first peak contracts slightly with the increase of the temperature (Figure 3a). It is worth mentioning that in the $g_{LiO}\left(r\right)$ curve at 400 K, the first, second, and third nearest neighbor Li–O distances are still well defined, whereas at higher temperatures, the peaks are smeared out, indicating a higher dynamics of the Li atoms. This demonstrates that octahedral coordination around the Li atoms is not regular any more above 600 K. A similar situation was found in experiments by Hou and Kirkpatrick [23], where the decomposition of LiAl-LDH began at temperatures slightly below 573 K.

For a more detailed analysis of the changes of the Li and Al atomic positions, we calculated the partial coordination numbers (PCNs), $N_{ij}$, which give the average number of *j*-type atoms within the cut-off distance, *R*_cut_, of an *i*-type atom from the PDF according to:(2)$$N_{ij}=\int_{0}^{R_{cut}}4\pi r^{2}\rho g_{ij}\left(r\right)dr,$$
where ρ is the number density [26,27]. The cut-off distance, *R_cut_*, was selected to be the distance corresponding to the tail of the first peak in the PDF.

Figure 4 shows the evolution of the PCN with an increasing temperature for Li–O and Al–O. For temperatures below 400 K, *N*_LiO_ is 6, which indicates that Li is octahedrally coordinated with O atoms. At temperatures above 450 K, *N*_LiO_ starts to decrease, suggesting that the Li atoms do not have an octahedral coordination any more, which may also be caused by a partial delithiation from the positively charged layers. Above 650 K, *N*_LiO_ stops decreasing and fluctuates around 2 with the further increasing temperature.

The evolution of the Al–O coordination environment with the temperature is quite different from the Li–O case. From the $g_{AlO}\left(r\right)$ shown in Figure 3b, it can be clearly seen that $g_{AlO}\left(r\right)$ shows a well-ordered structure even up to 1000 K. This result indicates that within the time-scale of MD simulations, the oxygen octahedron around Al is stable, which can also be seen from the PCN versus temperature curve, as presented in Figure 4. At temperatures above 1100 K, *N*_AlO_ starts to drop, which means that after the breakdown of the octahedral Li coordination above 450 K, the framework made up of the AlO_6_ octahedra remains stable until 1100 K. Above 1300 K, the octahedral coordination around Al becomes unstable.

To further study the different behaviors of Li and Al in LiAl_2_(OH)_6_Cl during the heating process, the mean square displacement (MSD) of Li and Al at 500 K were calculated and are presented in Figure 5. MSD is defined as [28]:(3)MSD=1N<∑i=1N|ric(t)−ric(0)|2>,
where *N* represents the number of type-*c* atoms in the system, while <A> denotes the ensemble average of a physical quantity, A. As it is known that the timescale of the Li diffusion process is of the order of picoseconds, it can be analyzed by MD simulations. It can be seen from Figure 5 that the Li atoms show a larger dynamics than the Al atoms. This can be explained by the heavier mass and higher charge of Al compared to Li. Within larger time intervals, the values of MSD increase with time as observed in [28]. In many kinds of lithium superionic conductors, e.g., Li_3_Y(PSO_4_)_2_, Li_5_PSO_4_Cl_2_ [29], and Li_7_P_3_S_11_ [30], an open site due to Li^+^ diffusion is found to be occupied by another Li^+^ within 10 ps.

From the MSD vs. *t* curve, the diffusion coefficient, *D*, can be calculated as follows:(4)$$D_{c}=\frac{1}{2d}\underset{t\to \infty }{\lim}\left(\frac{d}{dt}MSD\right),$$
(5)$$D_{c}=\frac{1}{2d}\left(\frac{MSD}{t}\right),$$
where d indicates the dimension of the system [28,29]. Equation (4) reduces to Equation (5) if the time dependence of MSD is approximately linear, which holds true in most cases. Thus, Equation (5) is also widely used in the literature to calculate the diffusion coefficient [29,31]. The calculated value of the Li^+^ diffusion coefficient by using Equation (5) is $3.13\times 10^{-5}cm^{2}/s$, which is higher than that of the Li^+^ ion calculated for Li_5_YPS_4_Cl_2_ (~$2.5\times 10^{-6}cm^{2}/s$) at 500 K [29]. The Li ion conductivity can be estimated by the Nernst–Einstein relation [29]:(6)$$\sigma =\frac{\rho z^{2}F^{2}D}{RT},$$
where ρ, *R*, and *F* represent the molar density of Li in the unit cell, gas constant (8.314 J∙mol^−1^∙K^−1^), and the Faraday’s constant (96,485 C∙mol^−1^), respectively. *z* is the charge number of Li, *z* = +1. According to the MSD, the Li conductivity for the current simulated system was calculated to be 708 mS/cm at 500 K.

Among several possibilities, we chose two for the movements of the Li atoms, i.e., within the layers or between the layers in LiAl_2_(OH)_6_Cl, see Figure 6, which we found reasonable from the structural point of view. The calculated structural energy is plotted against the diffusion coordinates, which shows an energy barrier of about 202 meV for the intra-layer diffusion within the Li–Al layers and 177 meV for the inter-layer diffusion path. Such low energy barriers may give rise to a high Li conductivity of partially delithiated Li_1-x_Al_2_(OH)_6_Cl_1-x_ (0 < x < 1). Although the Li conductivity may not be the only factor that determines a battery performance, its high value strongly recommends the possibility of utilizing this compound as a lithium superionic conductor.

## 3. Materials and Methods

All calculations were performed with the Vienna ab initio simulation package [32,33] (VASP, Fakultät für Physik of Universität Wien, Wien, Austria) and the Perdew–Burke–Ernzerhof (PBE) type generalized gradient approximation [34] (GGA) was applied to describe the exchange-correlation energy. We used a projector-augmented-wave method [32] for the ionic pseudo-potentials. Monkhorst-Pack [35] meshes of 4 × 4 × 4 was used to sample the reciprocal space. The energy cutoff was set to be 700 eV for both LiAl_2_(OH)_6_Cl and *γ*-Al(OH)_3_. These settings ensure the calculated total energy of the system converges to 1 meV. The energy cutoff and convergence criteria for the energy and force were set to be 1 × 10^−5^ eV and 0.001 eV/Å, respectively. The bonding properties were analyzed by using the LOBSTER code with the crystal orbital Hamilton population (COHP) [36,37,38,39]. The basis set of the employed pseudopotentials of H, O, Al, Cl, and Li are 1s^1^, 2s^2^2p^4^, 3s^2^3p^1^, 3s^2^3p^5^, and 2s^1^. For AIMD simulations, only the Г point was used to sample the Brillouin zone of the 2 × 2 × 1 supercell containing 128 atoms. The simulation was conducted in a canonical ensemble with a Nosé thermostat to control the temperature. Newton’s equations of motion were integrated using the Verlet algorithm with a time step of 1 fs in a simulation range of 10 ps.

## 4. Conclusions

In this work, the atomic structure and electronic structure of LiAl_2_(OH)_6_Cl were studied based on the first principles calculations. LiAl_2_(OH)_6_Cl was shown to be a wide-gap insulator with a calculated band gap of around 5 eV. The results from the AIMD simulation show that a delithiation process starts to occur and leads to the formation of Li_1-x_Al_2_(OH)_6_Cl_1-x_ (0 < x < 1) at temperatures higher than ~450 K. As a result, the Li^+^ ions start to become mobile as inferred from the change of the partial coordination number (PCN), whereas the Al–O coordination was stable until 1100 K. Based on the structure of LiAl_2_(OH)_6_Cl, the two most probable pathways for the Li diffusion were chosen and the analysis of the calculated results lead to two low energy barriers, one within the LiAl_2_(OH)_6_ layers and the other between such layers. The calculated large diffusion coefficient, $3.13\times 10^{-5}cm^{2}/s$, of the Li^+^ ion in the LiAl_2_(OH)_6_Cl material at 500 K suggests a high Li^+^ ion conductivity. The high mobility of the Li^+^ ions together with the high stability of the [Al_2_(OH)_6_] framework indicate that the title compound may be technically important as a superionic conductor at high temperatures.

## Figures and Tables

**Figure 1 molecules-24-02667-f001:**
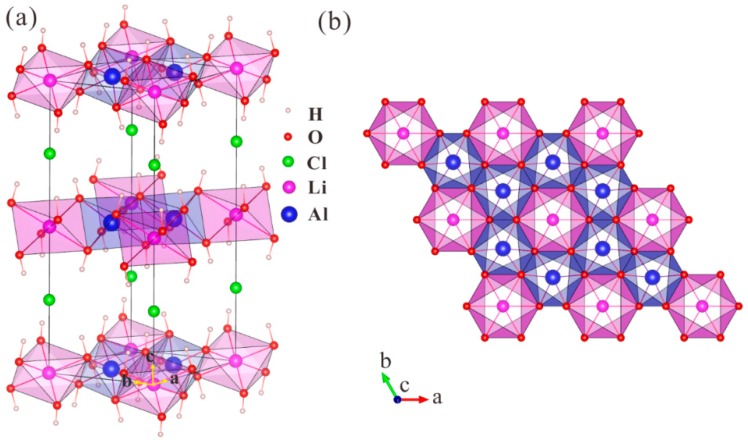
(**a**) Crystal structure of LiAl_2_(OH)_6_Cl with LiO_6_ and AlO_6_ octahedra indicated; (**b**) projection of a LiAl_2_O_6_ layer along the [001] direction.

**Figure 2 molecules-24-02667-f002:**
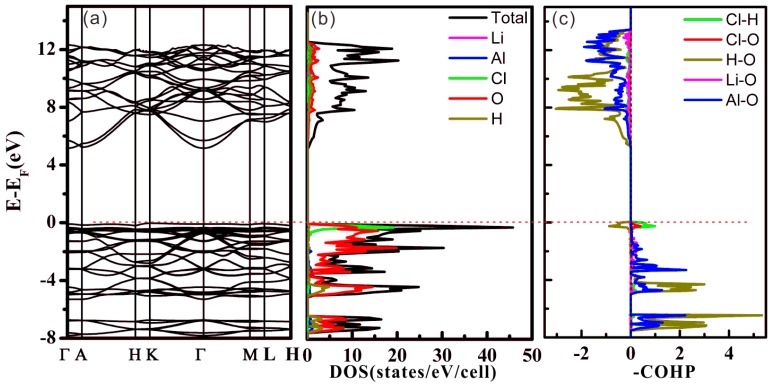
(**a**) Band structure, (**b**) density of states (DOSs), (**c**) crystal orbital Hamilton population (COHP) plots for LiAl_2_(OH)_6_Cl (the Li–O interactions are given in magenta, Al–O interactions in blue, H–O interactions in dark yellow, Cl–O interactions in red, Cl–H interactions in green.

**Figure 3 molecules-24-02667-f003:**
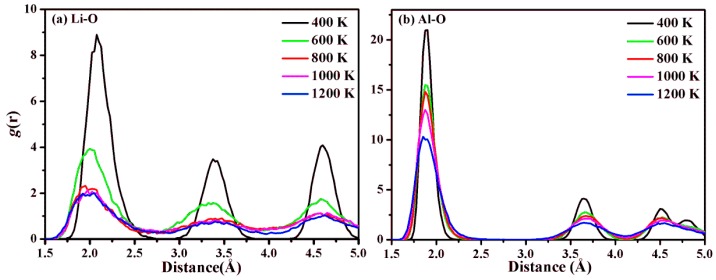
Pair distribution functions for (**a**) Li–O and (**b**) Al–O distances in LiAl_2_(OH)_6_Cl at various simulated temperatures.

**Figure 4 molecules-24-02667-f004:**
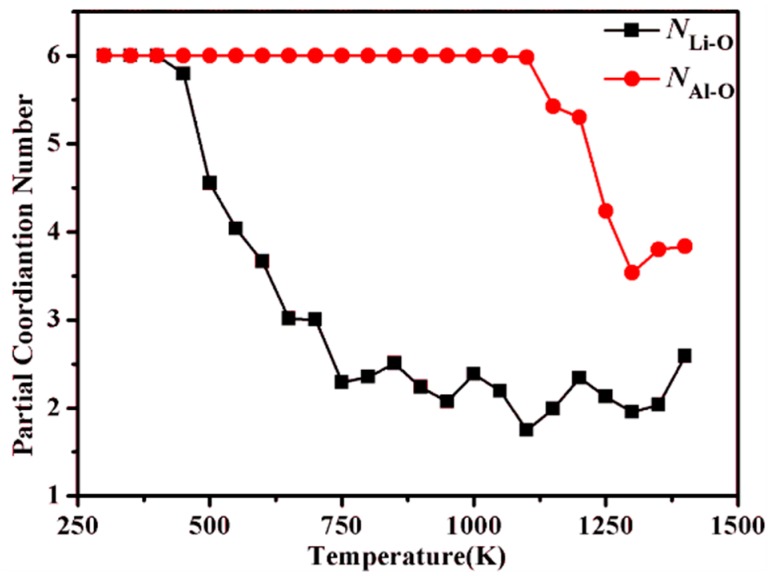
Evolution of the partial coordination numbers of Li and Al by O atoms with temperature in LiAl_2_(OH)_6_Cl.

**Figure 5 molecules-24-02667-f005:**
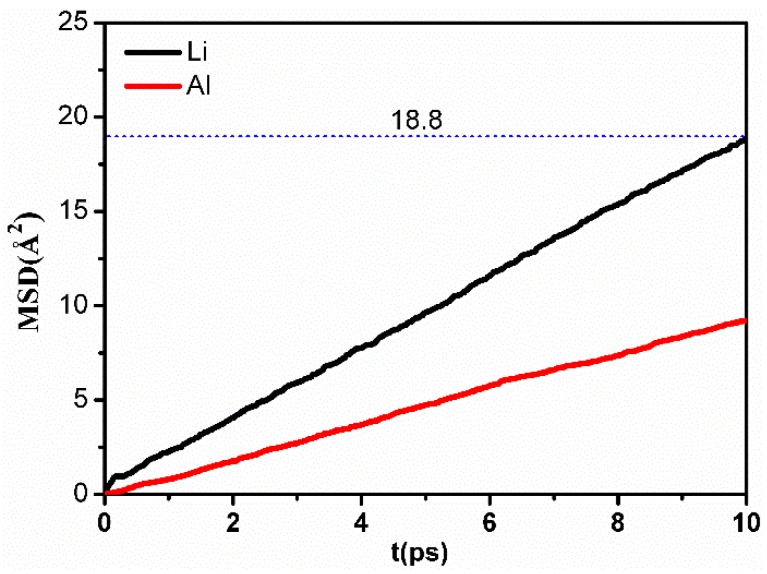
The mean square displacement (MSD) vs. time, *t*, for Li and Al in LiAl_2_(OH)_6_Cl at 500 K.

**Figure 6 molecules-24-02667-f006:**
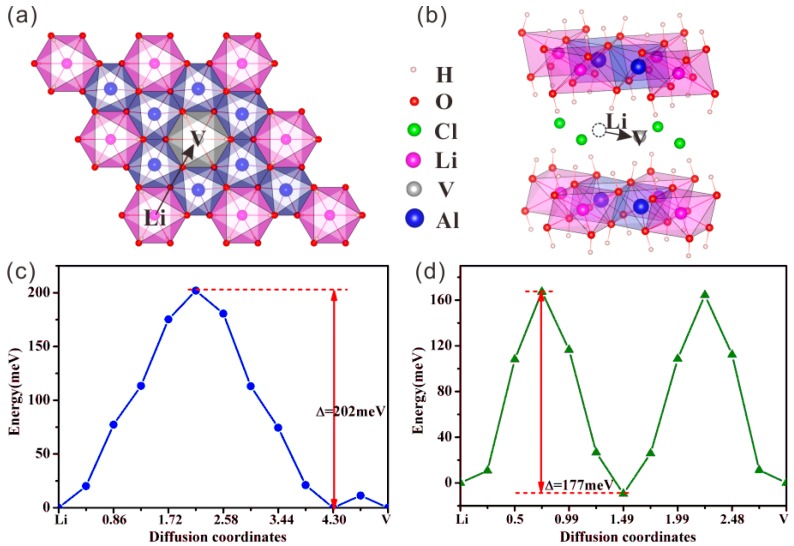
Lithium diffusion paths (**a**) within an LiAl_2_O_6_ layer and (**b**) in the region between the LiAl_2_O_6_ layers. The black arrows show the diffusion directions of the paths in the structure. ‘Li’ denotes the occupied site while ‘V’ denotes a vacant site. (**c**) and (**d**) are the energy profiles for these two paths, respectively.

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
