# Peer review of "Electronic Structure and Lithium Diffusion in LiAl_2_(OH)_6_Cl Studied by First Principle Calculations"

_molecules, 2019, doi:10.3390/molecules24142667_

Round 1
Reviewer 1 Report
The manuscript describes a set of calculations to determine the Li-diffusion properties of the layered double hydroxide LiAl2(OH)6Cl. Ab initio molecular dynamics (using density functional theory) has been used to analyse the coordination of Al and Li ions in the system as function of temperature, demonstrating that the coordination of Li reduces dramatically at temperatures above 450 K. The diffusion coefficient has been calculated from the mean square displacement of the Li ions, from which ion conductivities were estimated. Moreover, energetic barriers have been computed for Li diffusion along plausible pathways. The results show that the system is a potential Li superionic conductor. The methodology is sound, the results are sensible and the conclusions plausible. I can recommend the paper for publication, subject to the following minor comments:
1. In paragraph 3 of the introduction, the acronym 'DFT' is given without explanation.
2. In paragraph 2 of the results section, the calculated lattice parameters are said to be "slightly larger than the experimental ones". How much larger? It would be helpful to give the experimental values here. Do the experimental values correspond to room temperature?
3. On page 3, second paragraph: The percentage make up of the various bands are given with a high degree of precision. Are the authors sure that such precision is warranted? In the next paragraph, the acronym 'COHP' is given without explanation.
4. In section 2.2, the simulation times for the AIMD calculations at each temperature are not given.
5. Page 4, lines 131-132: "peak maximum ... remains almost the same with increasing temperature". This peak shifts considerably going from T = 400 K to T = 600 K.
6. Page 5, line 145: what number density is used? Is it the number density of Li ions, or ions in general?
7. Figure 5: "Square route" in the caption - should be "root". But are you plotting the mean square distance (as defined in equation 3) or the root mean square distance? In equation 3, what is N?
Reviewer 2 Report
I have reviewed the paper. These are my comments.
1) I think the authors should consult a professional editing service to render the English quality sufficient for publication.
2) The abstract and conclusion should be rewritten to reflect the important findings in the work.
3) Please point out in the introduction section why the Lithium Diffusion in LiAl2(OH)6Cl important and what is the application.Author Response
Please see the attachment.

Reviewer 3 Report
The authors analyse the Li-O and Al-O bonding in the LiAl2(OH)6Cl complex by applying AIMD calculations and calculating the RDF, partial coordination number and mean square displacement. The analysis sheds light on the detailed dynamics of the chemical bonds and is an interesting contribution to the literature. However, for this work to be considered for publication, the authors should state how they calculated the diffusion path in Figure 6, in addition to addressing the following comments:
- How did the authors obtain the lattice parameters for the LiAl2(OH)6Cl structure?
- On page 4, line 129, the authors state that "the peaks at ~3.65 Å and ~4.5 Å shift to larger values." However, it is not obvious that the peaks change at all. The authors need to clarify this change.
- The comparison between the diffusion coefficients on page 6 is problematic because the calculated D should be compared to the measured D in the "same" system at similar conditions. Comparing the theoretical D of LiAl2(OH)6Cl to the experimental D of a different compound does not build a proper argument.
Reviewer 4 Report
In this manuscript, first-principles calculations based on density functional theory were performed for Li-Al layered double hydroxides containing chlorine (LiAl-Cl-mH2O). First structural and electronic properties were obtained with the ground state calculations (T=0 K). Then finite temperature molecular dynamics simulations were performed to study the structural evolution of the LiAl-Cl-mH2O and the diffusivity of Li atom at high temperature.
–
I think the manuscript can be published in Molecules. However, following questions should be addressed before the publication.
Usually, the PBE functional (severely) overestimates interlayer distances (e.g. graphite) and dispersion correction is necessary. In the manuscript, on the contrary, it (lattice parameter c) is slightly underestimated. Can you elaborate the reason?
In the AIMD, the temperature was changed from 300 K to 1400 K with an increment of 50 K. At each temperature, how long the MD was performed? Was it long enough to equilibrate the system at that temperature?
In lines 100-103, the orbital lables are messed up, Cl 4s --> Cl 3s, etc.
In Figure 3 (b), there are two peaks between 4.5 A and 5.0 A at 400 K and the higher peak disappeared in higher temperatures. What is the origin of these peaks and why did one of them disappear at higher temperatures?
In lines 131-132, it is written that “gLiO(r), where the peak maximum at around 2.08 A remains almost the same with increasing temperature.” However, the peak seems to shift to the lower side with increasing temperature in Figure 3 (a).
At what temperature the energy along the diffusion paths shown in Figure 6 were calculated? 500 K?
Round 2
Reviewer 3 Report
The authors have responded reasonably to my minor comments. However, their response to my first comment, on Figure 6, is not satisfactory and will require a further revision.
The diffusion diagram in Figure 6, according to the description by the author, and as far as I have understood it, were calculated using single-point calculations. That is, the authors froze the Li atom at a specific point in space and did not relax the system at every point. This is not the correct method to obtain diffusion barriers. In fact, there are two possible methods to obtain such a barrier using VASP:
1- Nudged elastic band, which the author did not use
2- Constrained optimization (CO)
In the CO method, which the author attempted to use, as discussed above, one must perform a full relaxation of the whole system in order to avoid the introduction of unrealistic forces into the system. There are several resources on this in the literature which the authors can find out by googling.
And the authors should take this point into consideration: if you freeze the Li only, and let the rest of the atoms relax, you might get wrong outcomes, because the whole system might just shift with the Li atom.
In order to accept this work, the authors should clarify which method they have used in detail in the manuscript, and correct any methodological errors as discussed above, which might require updating Figure 6.